# Microstructure Evaluation of the Potential of Additive Manufactured Dissimilar Titanium–Aluminum Alloys

**DOI:** 10.3390/ma15249038

**Published:** 2022-12-17

**Authors:** Hideaki Nagamatsu, Takeyuki Abe, Hiroyuki Sasahara

**Affiliations:** 1Department of Mechanical Engineering and Intelligent Systems, The University of Electro-Communications, 1-5-1 Chofugaoka, Chofu-shi, Tokyo 182-8585, Japan; 2Graduate School of Science and Engineering, Saitama University, 255 Shimo-Okubo, Sakura-ku, Saitama-shi 338-8570, Japan; 3Department of Mechanical Systems Engineering, Tokyo University of Agriculture and Technology, 2-24-16 Naka-cho, Koganei-shi, Tokyo 184-8588, Japan

**Keywords:** additive manufacturing, wire and arc additive manufacturing, dissimilar metal deposition, titanium, aluminum, microstructure, input heat, penetration

## Abstract

Pure titanium (Ti) ERTi-2 was accumulated on an aluminum (Al) alloy ER5356 component via wire and arc additive manufacturing. The effect of processing parameters, mainly the input heat per unit length, on Ti/Al components was investigated. The microstructure of the Ti deposited layer and the Ti/Al reaction layer was analyzed using optical microscopy, scanning electron microscope, energy-dispersive spectroscopy, and an X-ray diffractometer. The fabrication of the surface layer equivalent to pure Ti as the used wire or Ti-Al alloy on the Al alloy components was achieved under low and high input heat conditions, respectively, although the Ti/Al components had low joinability and cracks at the reaction layer. Finally, the potential of additive-manufactured Ti/Al components with reference to our results and previous reports was discussed.

## 1. Introduction

Titanium (Ti) and aluminum (Al) dissimilar alloys have many advantages, such as being lightweight, being low-cost, and having high corrosion resistance. Ti/Al components have attracted the attention of aerospace and automotive industries due to their ability to decrease fuel consumption through lightweight construction. However, Ti and Al welded joints have been known to be difficult due to substantial differences in the chemical and physical characteristics. Moreover, many brittle Ti-Al intermetallic compounds (IMCs) are formed in the Ti/Al interface layer during the welding process, which significantly decrease the mechanical properties of the component. Some researchers have reported achieving high-strength joints of Ti and Al plates via laser welding [1,2,3], friction stir welding [4], magnetic pulse welding [5], and arc welding [6,7]. In the jointing of Ti/Al alloys, it is important to suppress IMCs’ formation to prevent molten pool formation and the dilution of two materials. For example, in laser welding, offsetting the laser beam towards the Al alloy material side suppresses IMCs formation at the Ti/Al interface.

Recently, there has been an increased interest in multi-material design via additive manufacturing (AM) technology, which is applicable to designs with a higher degree of freedom than jointing dissimilar metal plates. In particular, directed energy deposition (DED) is superior to other AM technologies for multi-material design because of its easy material switching and the possibility of selective material and heated energy supply [8,9]. Therefore, DED can potentially improve the dimensional limitations of dissimilar metal jointing and fabricate high-performance products with a Ti surface layer on an Al component.

However, it is difficult to suppress IMCs formation in the DED process because filler material and heat energy must supply the molten pool formed on the base material. Moreover, the accumulation of Ti layers on an Al component is much more difficult than the accumulation of Al layers on a Ti component because the deposition of molten Ti with a higher melting point than Al promotes the remelting of the Al component, i.e., IMCs formation. Tian et al. accumulated ER4043 layers on a TiAl6V4 wall or TiAl6V4 layers on an ER2319 wall by wire and arc additive manufacturing (WAAM) [10,11]. The thickness of the reaction layer and the tensile strength of the Ti on Al components were thicker and lower than those of the Al on Ti components, respectively. Hotz et al. manufactured functionally graded materials (FGMs) of TiAl6V4 and AlSi10Mg on AlMg3 substrate via laser-based DED [12]. FGMs with a step transition led to significantly less IMCs and cracks than a graded transition; however, they concluded that Ti deposition on Al alloy component was not advisable because both types of FGMs had a risk of premature component failure.

From the above, previous studies have not achieved high-strength joints of the Ti on Al components via AM and have mainly aimed at suppressing the Ti-Al IMCs formation. In addition, previous researchers have not focus on the fundamental effect of AM parameters on Ti/Al components and have just reported the experimental results in a very narrow process window. The experimental investigations in a wide process window are important to deepen understanding the potential of the additive manufactured Ti/Al components, even if the fabricated components have cracks or brittle IMCs. For example, the passive melting of Al via Ti deposition under low input heat can form the surface layer equivalent to the used Ti wire. In contract, the aggressive melting of Al via Ti deposition under high input heat has potential to form a Ti-Al alloy layers due to high dilution ratio [12,13]. However, nobody has revealed the effects of various processing parameters on the joinability and compositions between the Ti/Al interface.

Therefore, this paper describes the effects of various WAAM parameters, mainly focusing on the heat input per unit length, on the (1) dimension, (2) joinability, and (3) metallurgical structure of dissimilar Ti/Al components to clarify the applicability of fabricating Ti layers on an Al component for AM. Consequently, the condition ranges to obtain the Ti/Al components with characteristic joinability and metallurgical structure, which have not been clarified in previous studies, were revealed. Furthermore, solutions to accumulate a surface layer equivalent to Ti wire material or Ti-Al alloy on an Al component via DED including WAAM were proposed.

## 2. Materials and Methods

Table 1 and Table 2 show the nominal chemical compositions of the used wire and substrate materials. Ti and Al wires were ERTi-2 (WT2G, Daido Steel Co., Ltd., Aichi, Japan) and ER5356 (WEL MIG A5356-WY, NIPPON WELDING ROD Co., Ltd., Tokyo, Japan), respectively. The diameter of wires was 1.2 mm. The dimensions of Ti6Al4V and AlMg2.5 substrates were 150 mm × 100 mm × 5 mm and 150 mm × 150 mm × 5 mm, respectively.

The experimental setup is illustrated in Figure 1. A Ti bead on Ti plate, as shown in Figure 1a, was used to highlight the appearance features of Ti/Al dissimilar metal deposition. An Al bead or wall structure with 10–25 mm height was deposited onto an Al substrate using a metal inert gas (MIG) welding power source (P500L, DAIHEN Corp., Osaka, Japan). After the fabricated Al part cooled to room temperature, a Ti bead was deposited on the previous Al layer using cold metal transfer (CMT) welding source (TPS5000CMT, Fronius Co., Wels, Austria) as shown in Figure 2a. A welding robot (ARC Mate 100iC, FUNAC, Yamanashi, Japan) was employed to control the CMT welding torch. The local shield as shown in Figure 2b was attached to the CMT welding torch to prevent Ti beads from oxidation. This local shield was manufactured with reference to the paper by Bermingham et al. [14]. High purity argon (99.999%) flowed into the local shield through tow gas houses. The argon flow rate was 15 L/min through the CMT welding torch and 40 L/min through each of the two hoses. Table 3 shows the processing parameters. The heat input per unit bead length *Q* (J/mm) can be calculated from the following equation with the arc discharge current *I* (A), voltage *U* (V), and travel speed *TS* (mm/min).
*Q* = (*I*∙*U*∙60)/*TS*(1)

The metallographic samples were cut and polished from the WAAM components of Ti/Al dissimilar alloys at 50 mm from the welding start point. The polished surfaces of the specimens were etched in a mixture of 1 mL of hydrofluoric acid, 2 mL of acetic acid, and 21.5 mL of diluted water at room temperature for 2 min. The microstructure of the samples was observed via an optical microscope (VHX-6000, Keyence Corp., Osaka, Japan), a scanning electron microscope (SEM, VE-8800, Keyence Corp, Osaka, Japan), scanning direction of energy-dispersive spectroscopy (EDS, Element, EDAX LCC., California, America) and X-ray diffractometer (XRD, BRUKER D8 DISCOVER, AZ Science Co., Ltd., Nagano, Japan). The irradiation diameter in XRD was 0.5 mm. The international center for diffraction data (ICDD) to estimate IMCs is given in Table A1.

## 3. Results and Discussions

### 3.1. Appearance and Joinability of Ti/Al Components

Figure 3 shows the appearance and cross-sectional view of the WAAM components of Ti/Al dissimilar alloys. Dimension of Al alloy components and Ti deposition conditions affected the appearance of Ti/Al components and the penetration of the Al alloy side. Figure 3a,b show the results of the deposition of a Ti bead onto a Ti plate and an Al bead, respectively. Ti and Al beads had clear profiles, indicating that the Al beads were hardly molten. The Ti bead had an excessive side overhang, as shown in Figure 3b. In addition, the joint widths of the Ti/Al interface are smaller than the results of the deposition of the Ti bead on the Ti plate using the same processing conditions (*I* = 69 A, *U* = 17.3 V), as shown in Figure 3c. The molten Ti has high viscosity [15] and Al alloy as the base material has high thermal conductivity. Therefore, the molten pool during deposition Ti on Al was smaller than that during Ti on Ti, and sufficient joint widths could not be achieved.

Figure 3d–g show the results of the deposition of Ti on an Al wall structure. No. 9, shown in Figure 3d, had the same Ti deposition condition as No. 7, shown in Figure 3b; however, the upper layers were significantly melted from the center to the end. This was caused by the thermal diffusion of the Al alloy part as the base material. Generally, the thermal diffusion of the WAAM component is the highest during the deposition of a bead on a substrate, and the WAAM-fabricated component on the substrate is regarded as conductive of thermal resistance to the substrate [16,17]. In the process of deposition of Ti on an Al bead, the temperature of the deposited Ti bead did not rise easily because the heat added to the Al bead was quickly conducted to the substrate. In contrast, the Al wall structure significantly increased the conductive thermal resistance to the substrate. Therefore, the upper layer became hotter and eventually collapsed.

Ti beads using condition No. 10 (*I* = 69 A, *U* = 17.3 V, *Q* = 716 J/mm), with the same *I* and *U* and lower *Q* compared to No. 9, could be accumulated on the Al wall with little penetration. Meanwhile, conditions No. 11 and 12, shown in Figure 3f,g, had the same heat input (*Q* ≈ 1430 J/mm) but different *I*, *U*, and *TS*. With the increase in the current and voltage, Al wall parts of No. 11 and 12 melted significantly. Therefore, low input heat, current, and voltage provided Ti beads with little penetration, even into the Al wall structure with low thermal diffusion.

### 3.2. Fabrication of Pure Ti Layer Equivalent to the Used Wire on Al Alloy Component

Figure 4 shows the cross-sectional view and observation areas of deposition of Ti onto a single Al alloy bead using condition No. 7. Observation areas by SEM and EDS are illustrated by yellow dashed lines and a pink arrow, respectively.

Figure 4b shows the observed image of the Ti/Al interface (area 1−A). The spherical porosity formed at the Al bead side is caused by evolved and trapped hydrogen gas in the molten Al alloy during Al bead deposition [18]. According to Figure 4c, as shown in the EDS result along the pink arrow shown in Figure 4b, the deposited Ti and Al beads had component values equivalent to those of the wire material, respectively. The thickness of the IMC layer at the boundary was less than 10 µm. However, micro cracks occurred along the profile of the Al alloy bead due to thermal stress. In addition, Figure 4d shows that a void formed in the center of the Ti/Al interface (area 1−B). According to Figure 4e,f, as shown in the EDS result in area 1−B, Al elements were detected in the periphery of the voids at the Ti bead side. This is probably due to the detachment of the Al isles formed at the Ti side carried by direct liquid–liquid contact and material convection flow [3]. After solidification, Al isles presented at the Ti side were prone to fracture due to thermal stress and cracks. Finally, Al isles detached from the specimen surface during cutting the specimen, and a void formed.

### 3.3. Fabrication of Ti-Al Alloys Layer on Al Alloy Component

Figure 5a shows the cross-sectional view and observation areas of Ti deposited onto an Al wall structure using condition No. 9. Yellow and pink dashed rectangles illustrate SEM observation and EDS analyses areas, respectively. Light blue circles are the locations of XRD analyses. The pink arrow shown in Figure 5a illustrates the direction and the location of EDS multipoint analyses for the Ti element ratio of the fusion zone (FZ). The green dashed line illustrates the profile of the Al wall structure before Ti deposition. The upper layer of the Al wall was melted into the Ti deposition part. The IMC layers formed between the deposited Ti and FZ consisted of two layers with different elemental ratios at the Al wall and deposited Ti side. The total thickness of the IMC layers was approximately 1 mm. The IMC-FZ boundary was formed at a depth of approximately 4 mm from the top of the Al wall before Ti deposition. The fusion line was located at a depth of approximately 4 mm from IMC layers, and the Al wall formed a neck. Ti-Al IMCs with characteristic shapes were observed from the FZ to the Ti-deposited zone as the upper layer of the Al wall was melted into Ti deposited part. A schematic diagram of the characteristics and locations of IMCs is illustrated in Figure 5b, and the results of EDS and XRD analysis are summarized in Table 4.

Figure 5c shows a Ti agglomerate near the fusion line: isles of approximately 200 μm in diameter and needles approximately 50 μm in length (area 2−A). Chemical analysis of 2−A−1 revealed that the isles consisted of 43.3 at.% Ti and 54.0 at.%. The isles were scattered in FZ, and their dimensions increased toward the reversed building direction. Chemical compositions of the needles at 2−C−1 to −4 on area 2−C for the upper region of FZ shown in Figure 5d,e was 6.6–10.7 at.% Ti and 84.9–89.3 at.% Al. According to XRD at 2−a and the above EDS analysis results, the isle and the needle were TiAl_2_+TiAl and TiAl_3_, respectively (Figure 5h). FZ, except for the Ti agglomerate, was ER5356 from the EDS results at 2−A−2 and 2−B−1.

Figure 5g shows EDS multipoint analysis results. The pink arrow in Figure 5a indicates that the Ti element ratio on FZ decreases in the building direction. This is probably due to the difference in the solidification rate in the FZ. The solidification rate of the FZ near the fusion line is high because the FZ is cooled by the Al alloy wall with high thermal conductivity. Meanwhile, the solidification rate of the FZ near the IMC layers and FZ boundary decreased due to the Ti deposited at a high temperature. These indicate that the solidification of FZ occurred from the bottom to the top. First, TiAl_2_+TiAl, which has a high Ti element ratio and melting point, nucleated near the lower part of FZ with a high solidification rate. Then, the upper FZ gradually lacked Ti elements due to the nucleation of IMCs into the lower FZ. Finally, the needles of TiAl_3_, with a low Ti element ratio, were mostly formed in the upper FZ.

The IMC layer with a high ratio of Al or Ti element, as shown in Figure 5d, was formed on the Al wall or the deposited Ti side, respectively. Chemical analyses of 2−B−2 and 2−B−3 revealed that the IMC layers at the Al wall and deposited Ti side consisted of 29.3 at.% Ti and 68.1 at.% Al, and 41.0 at.% Ti and 57.2 at.% Al. According to XRD at 2−b and the above EDS results, the IMC layers at the Al wall and deposited Ti side were TiAl_2_ and TiAl_2_+TiAl, respectively (Figure 5i). The peak reaction of TiAl_3_, as shown in Figure 5h, was probably due to the needles of IMC that formed and surrounded IMC layers.

The deposited Ti layer was Ti-Al alloys, not equivalent to ER-Ti2 wire. Most of the deposited Ti had a uniform chemical composition: 51.3–51.9 at.% Ti and 46.3–46.8 at.% Al (areas 2−B−4 and 2−E). The deposited Ti consisted of TiAl+Ti_3_Al, as shown by XRD analysis of 2−e in Figure 5k. This is due to the diffusion of Al atoms into the molten Ti by the large remelting of the Al component. There were many cracks on the deposited Ti. In addition, a molten Al alloy was formed in the periphery of the deposited Ti, as shown in area 2−D (Figure 5f). The explanation for this is that the thermo-convective flow pushed semi-liquid Al into the molten Ti. Figure 5k shows that the molten Al alloy was equivalent to ER5356 as the filler material, and the needles formed in the molten Al alloy were TiAl_3_.

Finally, there is the possibility of additively manufactured dissimilar Ti/Al alloys, not only WAAM. Although there are still some problems, such as small joint width or cracks at the Ti/Al interface, it was found that a Ti wire equivalent layer and Ti-Al alloy layers can be accumulated on the Al component by applying low and high heat input conditions, respectively. Reducing thermal stress to suppress cracking involves applying induction heating (IH), which assists external heat input [19,20]. Bai et al. achieved more homogeneous heat input and lower residual stresses in WAAM components using IH. Improvement in the joint width between the Ti/Al interface is important to suppress oxidation in Al material [21] or the formation of an alumina film between Al and Ti [22]. Coating a fluor-based antioxidant flux on the Al component before depositing Ti is considered effective for ensuring the wetting of molten Ti when using alumina film [22]. Common key factors for creating pure Ti or Ti-Al alloy layers on Al alloy components are the control of the heat input and penetration. Therefore, tungsten inert gas (TIG) welding- or laser-based AM, which allow the independent control of heat input and material supply, could help to deposit dissimilar Ti and Al alloys. Cold spraying, which can significantly suppress the heat input [23,24], and control of the arc generation position by TIG welding seem to be effective in forming a Ti layer equivalent to the wire material. Rodrigues et al. developed a new WAAM method, namely, Ultracold-WAAM, in which the arc is generated primarily on the wire side rather than the base metal, achieving lower temperatures in the WAAM components and lower dilution of the preceding layer [25]. Meanwhile, it is important when creating a Ti-Al alloy layer to create deep penetration on the Al component. Therefore, increasing the thermal energy concentration via coating active flux [26] or a laser heat source, not only parameters with high heat input, is effective for increasing the penetration. Moreover, DED can create alloy layers with desired metal compositions by simultaneously feeding multiple materials [13,27,28,29]. Therefore, there is a possibility of the deposition of a Ti-Al alloy layer with desired compositions by improving the prediction of the dilution ratio in AM when feeding multiple materials. Thus, our experimental results and the methods proposed by previous studies are expected to further advance additively manufactured dissimilar Ti and Al alloys. In future studies, the Ti-bead formation process will be observed. The influences of coating a flux and WAAM parameters on behavior of molten pool and the joinability between the Ti/Al interface will be investigated using a high-speed camera to deepen understanding the phenomenon of dissimilar Ti/Al alloys deposition.

## 4. Conclusions

In this study, the experiments on WAAM fabrication of Ti on Al were conducted to clarify the possibility of fabricating a surface layer equivalent to pure Ti wire or Ti-Al alloy on Al alloy components. The following conclusions could be drawn:

The surface layer equivalent to pure Ti wire could be formed on an Al component under low heat input conditions. The Ti beads had a large side overhang and low penetration. The thickness of the IMC layer formed between the Ti/Al interface was less than 10 µm due to the low dilution ratio. The joint width at the Ti/Al boundary was narrow due to the low wettability of Ti beads.

The surface layer equivalent to TiAl+Ti_3_Al could be formed on an Al component under high heat input conditions. The boundary layer between the Ti/Al dissimilar alloys was approximately 1 mm in thickness, and FZ reached a depth of approximately 4 mm from the boundary layer due to the large penetration and dilution ratio. The boundary layer consisted of two IMC layers, TiAl+TiAl_2_ and TiAl. TiAl+TiAl_2_ with isles and TiAl_3_ with needles were observed in the FZ.

It Is important for dissimilar Ti and Al alloy deposition to control the input heat, suppress cracks, and improve the wettability of molten Ti. Therefore, heat sources such as TIG welding, thermal stress relief by IH, and the removal of alumina film are considered effective for controlling the input heat, suppressing cracks, and improving wettability, respectively.

## Figures and Tables

**Figure 1 materials-15-09038-f001:**
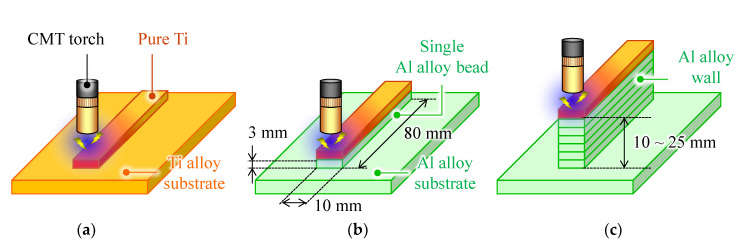
Schematic of deposition processes for Ti bead onto (**a**) Ti6Al4V substrate, (**b**) an Al alloy bead, and (**c**) Al wall structure.

**Figure 2 materials-15-09038-f002:**
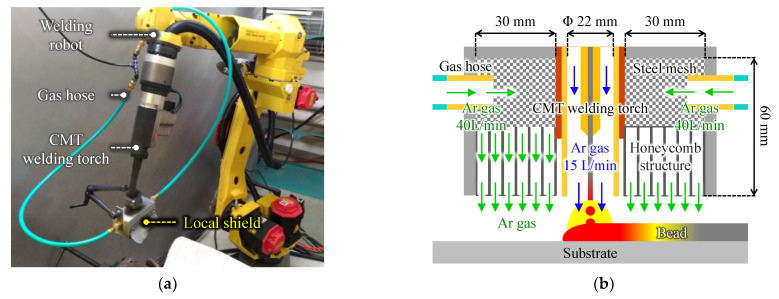
(**a**) Wire and arc additive manufacturing machine and (**b**) internal structure of the local shield.

**Figure 3 materials-15-09038-f003:**
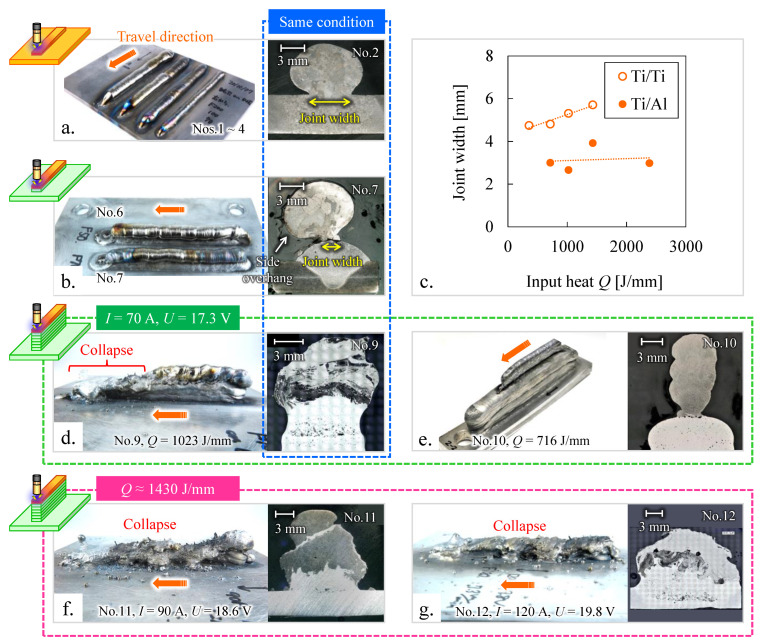
Appearance and cross-sectional views of Ti/Ti or Ti/Al components: Ti beads deposited onto (**a**) Ti substrate and (**b**) single Al alloy bead. (**c**) Comparison of Ti/Ti and Ti/Al joint widths. Ti beads deposited onto Al wall structures with the conditions of (**d**) No. 9 the same as No. 2 and 7; (**e**) No. 10 the same *I* and *U*, and lower *Q* than No. 9; (**f**) No. 11 with higher *I*, *U*, and *Q* than No. 9; and (**g**) No. 12 with higher *I* and *U* and same *Q* as No. 11.

**Figure 4 materials-15-09038-f004:**
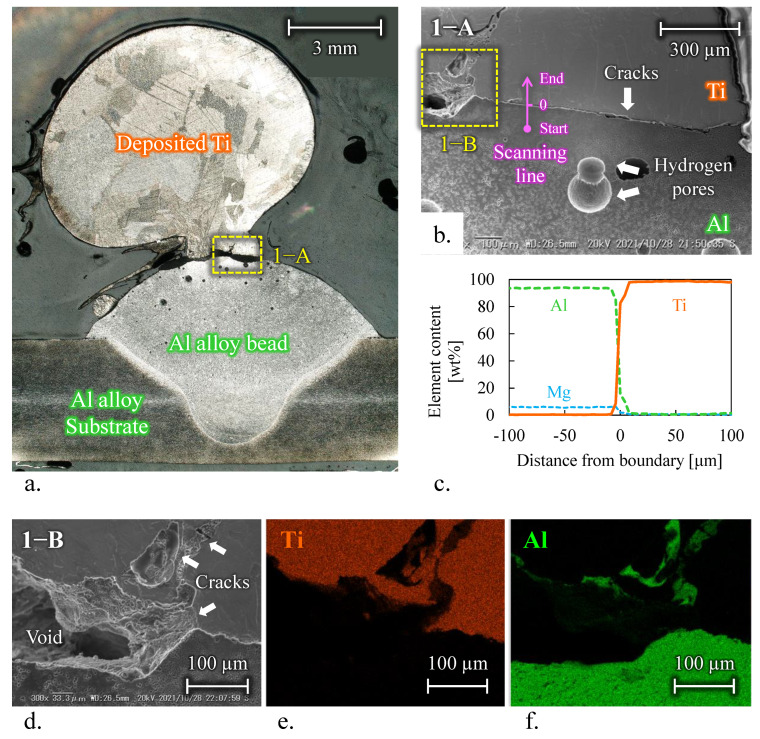
Ti onto single Al alloy bead using fabrication condition No. 7; (**a**) Overview. (**b**) Area 1−A: Ti/Al interface. (**c**) Result of EDS-linear analysis. (**d**) Area 2−B: void and cracks on Ti/Al interface. EDS map of area 2−B corresponding to (**e**) Ti and (**f**) Al.

**Figure 5 materials-15-09038-f005:**
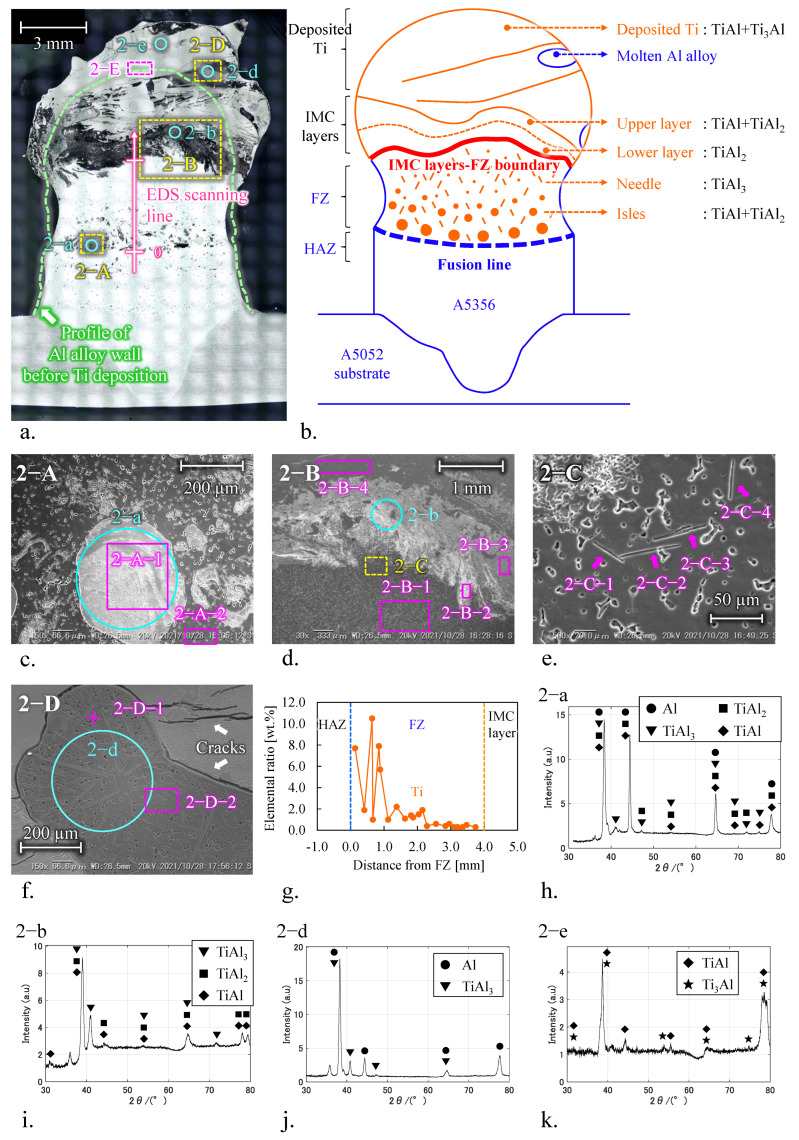
Ti deposited onto Al alloy wall using condition No. 9. (**a**) Overview. (**b**) Schematic diagram to illustrate the locations of the IMCs and the boundaries. (**c**) Area 2−A: isles IMC above the boundary between FZ and HAZ. (**d**) Area 2−B: intermetallic layers between FZ and Ti-deposited part. (**e**) Area 2−C: needle IMC below the intermetallic layer. (**f**) Area 2−D: molten Al alloy outside of Ti-deposited part. (**g**) Results of EDS analysis for Ti element on FZ along building direction. XRD analyses of (**h**) isles IMC on 2−a, (**i**) Ti-side interlayer on 2−b, (**j**) molten Al alloy on 2−d, and (**k**) the top of Ti-deposited part on 2−e.

**Table 1 materials-15-09038-t001:** Nominal chemical composition of the used Ti wire and substrate material (wt.%).

Material	Dimension	O	H	N	C	Fe	Al	V	Ti
ERTi-2	Wire	≤0.15	≤0.008	≤0.02	≤0.03	≤0.20	-	-	Bal.
Ti6Al4V	Plate	≤0.20	≤0.015	≤0.05	≤0.08	≤0.30	5.50–6.75	3.50–4.50	Bal.

**Table 2 materials-15-09038-t002:** Nominal chemical composition of the used Al wire and substrate material (wt.%).

Material	Dimension	Si	Fe	Cu	Mn	Cr	Zn	Mg	Al
ER5356	Wire	≤0.25	≤0.40	≤0.10	0.05–0.20	0.05–0.20	-	4.5–5.5	Bal.
AlMg2.5	Plate	-	≤0.40	≤0.10	≤0.10	0.15–0.35	≤ 0.10	-	Bal.

**Table 3 materials-15-09038-t003:** Processing parameters for Ti deposition.

Test No.	Previous Layer		Current	Voltage	Travel Speed	Wire Feed Speed	Input Heat	Total Flow Rate Ar Gas
	Material	Dimension	*I*	*U*	*TS*	*WFS*	*Q*	
	-	-	A	V	mm/min	m/min	J/min	L/min
1	Ti6Al4V	Plate	69	17.3	50	3.6	1432	95
2					70		1023	
3					100		716	
4					200		358	
5	ER5356	Singlebead	69	17.3	30	3.6	2387	95
6					50		1432	
7					70		1023	
8					100		716	
9	ER5356	Wall	69	17.3	70	3.6	1023	95
10			69	17.3	100	3.6	716	
11			90	18.6	70	5.5	1435	
12			120	19.8	100	7.3	1426	

**Table 4 materials-15-09038-t004:** EDS results of the selected areas (at.%).

Location	Characteristic	Ti	Al	Mg	Possible Phase
2−A−1	Isles	43.3	54.0	2.7	TiAl_2_ + TiAl
2−A−2	Al alloy wall	0.0	95.3	4.7	Al
2−B−1	Al alloy wall	0.2	94.1	5.6	Al
2−B−2	Al alloy-side interlayer	29.3	68.1	2.6	TiAl_2_
2−B−3	Ti-side interlayer	41.0	57.2	1.8	TiAl_2_ + TiAl
2−B−4	Ti deposited part	51.9	46.4	1.7	TiAl + Ti_3_Al
2−C−1,2,3,4	Needle	6.6–10.7	84.9–89.3	3.7–4.8	TiAl_3_
2−D−1	Needle	11.9	83.2	4.9	TiAl_3_
2−D−2	Molten Al alloy	0.1	93.8	6.0	Al
2−E	Ti-deposited part	51.3	46.8	1.9	TiAl + Ti_3_Al

## Data Availability

Not applicable.

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
