# Peer review of "Microstructure Evaluation of the Potential of Additive Manufactured Dissimilar Titanium–Aluminum Alloys"

_materials, 2022, doi:10.3390/ma15249038_

Round 1
Reviewer 1 Report
The use of abbreviations in the abstract section may be avoided.Author Response
Thank you for your valuable comments. And I applogize for my late revisions, regardless of your quick response. We responsed as follows.
Point 1: The use of abbreviations in the abstract section may be avoided.
Response 1:
I have reivised the all abbreviations except “Ti” and “Al” in the abstract.

Reviewer 2 Report
The authors have conducted experiments with cladding titanium over aluminum, titanium over titanium, and titanium over an aluminum wall. The authors have not mentioned the significance of such experiments. As it is very difficult to deposit titanium over aluminum, as mentioned in much of the literature, the experiment seems redundant. Ti over Ti also has plenty of metallurgical defects as we study in the literature. Making a 3D component using Ti beads over Ti is also a difficult task. The industrial application may be limited because of such defects and difficulties faced during the process. However, the experiments have been conducted well and characterized for their metallurgical properties. The authors have also concluded the paper well. Therefore, the paper may be considered for publication, provided, the authors explain the reason for conducting the redundant experiments for concluding that the combinations such as Ti over Ti or Ti over Al will not work.
Author Response
Thank you for your valuable comments. And I applogize for my late revisions, regardless of your quick response. We responsed as follows.
Point 1: The authors have conducted experiments with cladding titanium over aluminum, titanium over titanium, and titanium over an aluminum wall. The authors have not mentioned the significance of such experiments. As it is very difficult to deposit titanium over aluminum, as mentioned in much of the literature, the experiment seems redundant. Ti over Ti also has plenty of metallurgical defects as we study in the literature. Making a 3D component using Ti beads over Ti is also a difficult task. The industrial application may be limited because of such defects and difficulties faced during the process. However, the experiments have been conducted well and characterized for their metallurgical properties. The authors have also concluded the paper well. Therefore, the paper may be considered for publication, provided, the authors explain the reason for conducting the redundant experiments for concluding that the combinations such as Ti over Ti or Ti over Al will not work.
Response 1:
I have reivised mainly the latter half of the introduction from line 50 to line 77 to highlight the unresolved issues in previous studies and the novelty of this study. The unresolved issues in previous studies are a narrow process window. As far as I know, the previous studies about dissimilar Ti-Al alloys have described the results of the fabricating tests using a single combination of parameters. They may have wanted to show only the best or most characteristic data because dissimilar Ti-Al alloys deposition are extermely difficult and cannot be succesufuly bonded under most conditions. However, the experimental investigations in a wide process window are important to deepen understanding the potential of the additive manufactured Ti/Al components, even if the fabricated Ti/Al components have cracks or low joinability. In addition, we did not invesitigate in this study; however, Ti-bead formation process – dloplet transfer and behavior of molten pool – is also important to deepen understanding the phenomenon of dissimilar Ti/Al alloys deposition. These tasks will be conducted in the future and have been added from line 268 to line 271). Futhermore, the issues of dissimilar Ti-Al alloys deposition, formation of brritle IMCs or cracks, are different from the issues of simmilar Ti alloys deposition. Therefore, our propose, investigation of the effects of various WAAM parameters on the Ti/Al components, have not been clarified in previous studies and is beneficial.
Reviewer 3 Report
The paper entitled "Microstructure Evaluation of the potential of Additive manufactured Dissimilar Titanium–Aluminum alloys" presents an interesting work in the performance of WAAM for dissimilar manufacturing of Ti-Al structures. From my point of view, the topic is of great interest and the quality is good. Only some comments.
· In the introduction please end with the novelty of your work not with reference to prior works. Add reference to recent works on the same field:
o https://doi.org/10.3390/ma15175828
o https://doi.org/10.1016/j.matdes.2020.109370
o https://doi.org/10.1016/j.cirpj.2022.06.018
· Please avoid the use of 1st person verbs: “Furthermore, we propose solutions to accumulate”
· Please placed the actual set-up
· The conclusions seem adequate although it does not appear that the desired result has been found.
· Please highlight future lines of research.
From my point of view the result is of great interest. The fabrication of dissimilar walls with Ti and Al welding is of great difficulty. The observed results do not solve the problem but give important guidelines for the future and I suggest their publication.
Author Response
Thank you for your valuable comments. And I applogize for my late revisions, regardless of your quick response. We responsed as follows.
Point 1: In the introduction please end with the novelty of your work not with reference to prior works. Add reference to recent works on the same field:
- https://doi.org/10.3390/ma15175828
- https://doi.org/10.1016/j.matdes.2020.109370
- https://doi.org/10.1016/j.cirpj.2022.06.018
Response 1:
I have reivised mainly the latter half of the introduction from line 50 to line 77 to highlight the unresolved issues in previous studies and the novelty of this study. In addition, I have added the refferences regarding the reviews of multi-materialization via DED on line 42. Thank you for the interested references you recommend; however, our paper has focused on dissimilar Ti-Al alloys deposition so that I did not cite these references. I have also investigated for similar Al alloy depositon and dissimilar ferrous metals deposition so that I will cite your recommendations when writing these papers.
Point 2: Please avoid the use of 1st person verbs: “Furthermore, we propose solutions to accumulate”.
Response 2:
I have avoided the use of 1st person verbs.
Point 3: Please placed the actual set-up.
Response 3:
I have added Figure 2 (a, b) and expression about the actual set-up from line 89 to line100.
Point 4: Please highlight future lines of research.
Response 4:
I have added our future tasks from line 268 to line 271. In this paper, we have just investigated about the relationship between the various WAAM parameters and the joinability and compositions between the Ti/Al interface. The Ti-bead formation process – dloplet transfer and behavior of molten pool – is also important to deepen understanding the phenomenon of dissimilar Ti/Al alloys deposition. Thus, we will investigate the influences of coating a flux and WAAM parameters on behavior of molten pool and the joinability between the Ti/Al interface.
Round 2
Reviewer 2 Report
The authors have justified the experimentation process and have explained that there are a few more studies that will be carried out in the future to find out the associated problems described in the previous review.